# Spatially inhomogeneous competition between superconductivity and the charge density wave in YBa$_2$Cu$_3$O$_{6.67}$

J. Choi[1], O. Ivashko[1,2], E. Blackburn[3,4], R. Liang[5,6], D.A. Bonn[5,6], W.N. Hardy[5,6], A.T. Holmes [7], N.B. Christensen[8], M. Hücker[9], S. Gerber [10], O. Gutowski[2], U. Rütt [2], M.v. Zimmermann [2], E.M. Forgan[3], S.M. Hayden[11] & J. Chang [1]

The charge density wave in the high-temperature superconductor YBa$_2$Cu$_3$O$_{7-x}$ (YBCO) has two different ordering tendencies differentiated by their *c*-axis correlations. These correspond to ferro- (F-CDW) and antiferro- (AF-CDW) couplings between CDWs in neighbouring CuO$_2$ bilayers. This discovery has prompted several fundamental questions: how does superconductivity adjust to two competing orders and are either of these orders responsible for the electronic reconstruction? Here we use x-ray diffraction to study YBa$_2$Cu$_3$O$_{6.67}$ as a function of magnetic field and temperature. We show that regions with F-CDW correlations suppress superconductivity more strongly than those with AF-CDW correlations. This implies that an inhomogeneous superconducting state exists, in which some regions show a fragile form of superconductivity. By comparison of F-CDW and AF-CDW correlation lengths, it is concluded that F-CDW ordering is sufficiently long-range to modify the electronic structure. Our study thus suggests that F-CDW correlations impact both the superconducting and normal state properties of YBCO.

[1] Physik-Institut, Universität Zürich, Winterthurerstrasse 190, CH-8057 Zürich, Switzerland. [2] Deutsches Elektronen-Synchrotron DESY, 22607 Hamburg, Germany. [3] School of Physics and Astronomy, University of Birmingham, Birmingham B15 2TT, UK. [4] Division of Synchrotron Radiation Research, Department of Physics, Lund University, Sölvegatan 14, 22100 Lund, Sweden. [5] Department of Physics & Astronomy, University of British Columbia, Vancouver, Canada. [6] Canadian Institute for Advanced Research, Toronto, Canada. [7] European Spallation Source ERIC, Box 176, SE-221 00 Lund, Sweden. [8] Department of Physics, Technical University of Denmark, DK-2800 Kongens Lyngby, Denmark. [9] Department of Condensed Matter Physics, Weizmann Institute of Science, 7610001 Rehovot, Israel. [10] Laboratory for Micro and Nanotechnology, Paul Scherrer Institut, Forschungsstrasse 111, CH-5232 Villigen, PSI, Switzerland. [11] H. H. Wills Physics Laboratory, University of Bristol, Bristol BS8 1TL, UK. ✉email: S.Hayden@bristol.ac.uk; johan.chang@physik.uzh.ch

Many theories and experiments point to an inhomogeneous nature of the superconductivity of high-temperature cuprate superconductors (HTC). Indeed, inhomogeneity in the sense of phase separation, intertwining of competing order parameters, stripes or pair-density-waves[1–5] may be at the heart of cuprate superconductivity. Underdoped high-temperature superconductors exhibit competing tendencies towards charge-density-wave (CDW) and superconducting (SC) orders[6–10]. This means that the effects of quenched disorder may induce defects that disrupt or suppress the CDW order, and in turn lead to a resurgence of superconductivity. The application of a magnetic field provides another interesting control parameter because the introduction of vortices into the superconducting state tips the energy balance in favour of CDW order.

In this context, YBa$_2$Cu$_3$O$_{6+x}$ (YBCO) is an important model system where large onset temperatures of both superconductivity and CDW order are present[8,9,11,12]. As a function of both magnetic field and uniaxial stress, two intimately related CDW ordering tendencies have been realised[13–16]. These correspond to different ordering patterns along the c-axis[13,14], as schematically illustrated in Fig. 1. Magnetic field (or uniaxial stress) induces an in-phase ferro-coupled CDW (F-CDW) along the c-axis[14,15] on top of the original out-of-phase bi-axial antiferro-coupled CDW (AF-CDW) order. Quenched disorder, that in YBCO is naturally introduced through imperfections in the chain layer, may have important implications for the CDW order[15,17,18]. It is believed that disorder locally favours the AF-CDW order and that the evolution of the CDW correlations with magnetic field may be understood as a crossover transition in which the CDW coupling along the c-axis varies[14,18]. The introduction of normal vortices with field leads to a more ordered CDW. This suggests disorder associated with vortices is of secondary importance.

More recently, the case where competing tendencies towards CDW and SC order in the presence of topological defects due to quenched disorder and of vortices in the SC state has been considered[17]. It was demonstrated theoretically that a fragile SC state can occur at low temperatures and at high magnetic field. This state is based on regions in which the CDW order is weakened due to defects where locally superconducting halos can form. These regions can then couple to form a state with global SC phase coherence. Given the richness of theoretical possibilities when superconductivity competes with CDW order in the presence of vortices and weak disorder, it is of great interest to scrutinise the magnetic field-induced phase competition in YBCO.

Here we present a comprehensive study of the F-CDW and AF-CDW correlations in YBa$_2$Cu$_3$O$_{6.67}$. By varying the magnetic field strength B – applied along the crystallographic c-axis – and temperature T, new insights into in-plane correlation lengths in the phase space (B, T) and competition of the correlations with superconductivity are obtained. F-CDW correlations are traced from low fields. From the diffraction intensities of F- and AF-CDW correlations versus temperature and magnetic field, the competition with superconductivity is studied. In particular, we extract the characteristic temperatures below which the respective CDW correlations are suppressed. They are significantly different at high magnetic fields, where the F-CDW correlation length becomes large, and the F-CDW correlations are suppressed at a significantly lower temperature scale. These different temperatures indicate where superconducting order becomes strong enough to influence the F-CDW and AF-CDW correlations. This observation suggests that superconductivity in the F-CDW regions is more strongly suppressed by the CDW. In this fashion, inhomogeneous superconductivity emerges due to competition with CDW correlations which are locally different. Consequently, regions with weaker or more fragile superconductivity are created. Another implication of our study is that in-plane correlation lengths can be deduced across the (B, T) phase space. We extract the in-plane correlation lengths at the field-dependent temperature scales set by the phase competition between superconductivity and the two CDW ordering tendencies. It is demonstrated that the F-CDW in-plane correlation length exceeds that of the AF-CDW order down to magnetic fields as small as 5 T. This result raises the question as to whether F-CDW correlations may be related to the Fermi surface reconstruction. Together with the recent finding that uniaxial strain induces F-CDW order[16], our results suggest that F-CDW order in YBCO plays an important role for both the electronic structure and in rendering superconductivity into an inhomogeneous state that includes a more fragile flavour.

## Results

**Charge-density-waves in YBCO.** Charge density wave (CDW) ordering has been found to be a universal property of hole doped cuprates[6–9,19–21]. YBCO has a bilayer structure with relatively close (d = 3.3 Å) pairs of CuO$_2$ planes, separated by layers containing CuO chains running along the b-axis. In YBCO, the CDW results in weak scattering centred at $\mathbf{Q}_{CDW}(\ell) = \boldsymbol{\tau} + \mathbf{q}_{CDW} + \ell \mathbf{c}^\star$, where $\boldsymbol{\tau}$ is a reciprocal lattice point of the unmodulated structure and $\mathbf{q}_{CDW} = (\delta_a, 0, 0)$ or $(0, \delta_b, 0)$[8,9,11]. For the YBa$_2$Cu$_3$O$_{6.67}$

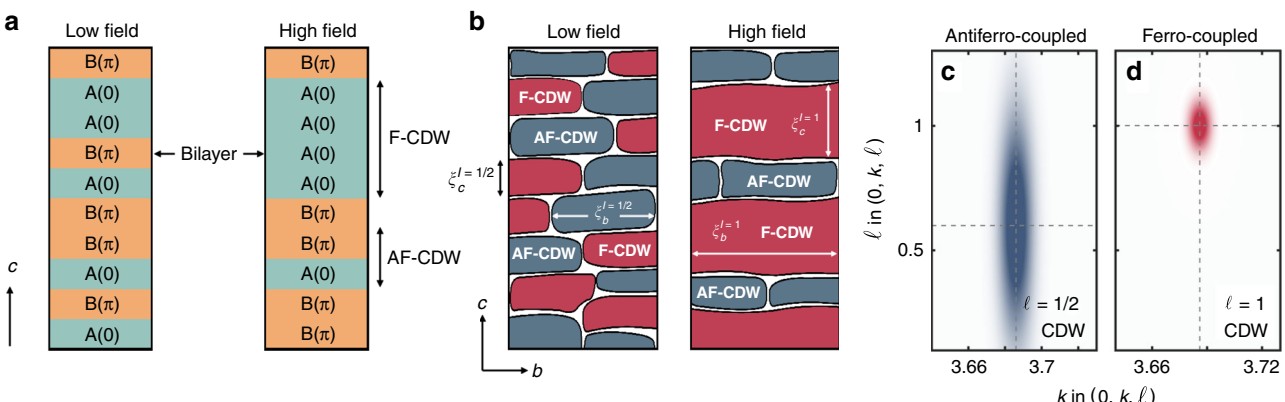

**Fig. 1 Schematic real-space illustration of the disordered charge density wave (CDW) in YBa$_2$Cu$_3$O$_{6.67}$. a** Representative CDW stacking sequences along the c-axis at low and high magnetic field for a fixed point in the a-b plane. A and B represent the two possible phases in the CDW modulation in each bilayer[14]. **b** Extension to the b-c plane. Spatially separated regions where the F-CDW ($\ell = 1$) and AF-CDW ($\ell = 1/2$) correlations in **a** are present. **c, d** Idealised diffraction intensity for the F-CDW and AF-CDW correlations at high field.

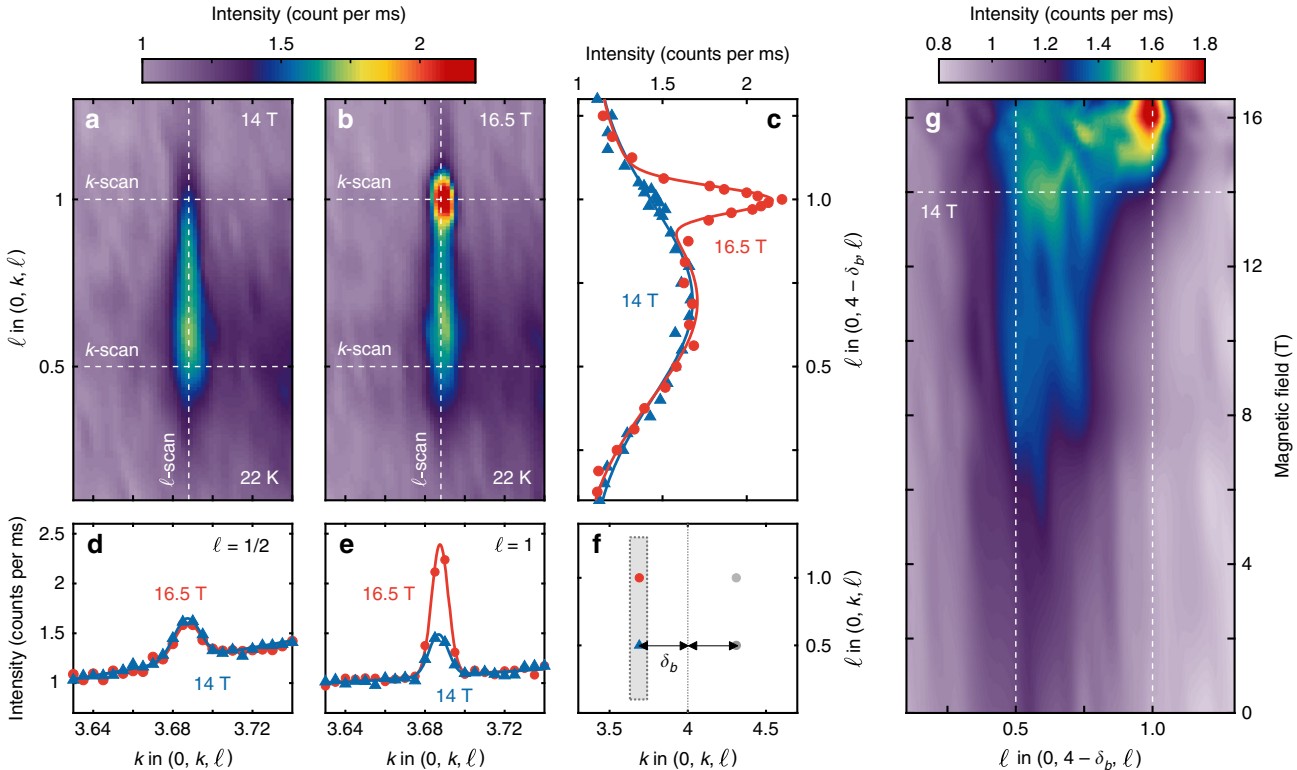

**Fig. 2 Magnetic-field-induced correlations and three-dimensional CDW order in YBa$_2$Cu$_3$O$_{6.67}$. a, b** X-ray diffraction intensity maps around $(0, 4 - \delta_b, \ell)$ with $B = 14$ T and 16.5 T, respectively. **c, d,** and **e** are $\ell$- and $k$-dependent intensity profiles along the vertical and horizontal white dashed lines in **a, b**. **f** Delineation of the scattering plane and probed reciprocal space (grey). The in-plane incommensurability $\delta_b$ is defined by the horizontal arrow. **g** Diffraction intensity map displayed in false colour, as a function of $\ell$ and magnetic field. Source data are provided as a Source Data file.

($T_c = 67$ K) composition studied here, two charge-density-wave components have been identified[13–15]. The first, found in zero magnetic field, develops below $T_{CDW} = 140(10)$ K with $\delta_a = 0.305$ (2) and $\delta_b = 0.314(2)$. The phase of this CDW in neighbouring bilayers has an anti-parallel correlation with only weak correlation along the $c$-axis, manifested by a broad intensity peak with an $\ell \approx 1/2$ modulation. The intensity of these rods of scattering is enhanced by the application of a magnetic field along the $c$-axis[9,22,23]. Figure 2a displays this rod in the $(0, k, \ell)$ scattering plane for $B = 14$ T. A second magnetic-field-induced CDW exists along the $b$-axis direction[13–15]. This CDW has the same phase across neighbouring bilayers (ferro-coupled) and hence is manifested by a $\ell = 1$ reflection on top of the $\ell \sim 1/2$ scattering rod (see Fig. 1c, d and Fig. 2a, b). In this paper we use the term "CDW order" loosely to mean static CDW correlations with a finite correlation length which develop at low temperatures and high magnetic fields.

Modelling[14] of the $c$-axis correlations in YBCO suggests that magnetic field changes the inter-bilayer (IB) coupling of the CDW. This is an indirect effect in that the field suppresses the superconductivity which changes the IB coupling. The CDW will also be pinned by defects in the chain layer. Thus, we can view the evolution of the CDW with field as a crossover transition. At low field $B \lesssim 14$ T, the CDW is pinned and the system displays weakly correlated $\ell = 1/2$ order. At very high field ($B \gg 14$ T), in-phase coupling of the bilayers is favoured, which is sufficient to overcome the pinning and forms a state with long range $l = 1$ order along the $b$-axis only.

**Two-component analysis.** The two CDW orderings can be analysed separately as a function of temperature and magnetic field. Here we employ three separately independent but consistent

methodologies for data analysis. First, as shown in Figs. 2d, e and 3d, e, the in-plane correlations and diffraction intensities can be inferred from Gaussian fits versus $\mathbf{Q}_b$ at $\ell = 1/2$ and $\ell = 1$. The correlation lengths are defined as $\xi_b^{\ell=1/2,1} = 1/\sigma_b^{\ell=1/2,1}$ where $\sigma_b^{\ell=1/2}$ and $\sigma_b^{\ell=1}$ are the respective standard deviations. Second, along the $(0, 4 - \delta_b, \ell)$ direction, the out-of-plane correlation lengths $\xi_c^{\ell=1/2,1}$ can be deduced by fitting with a double Gaussian function (Fig. 3c). The centre in $\ell$ of the broad AF-CDW signal rises slightly with magnetic field[14] (the values that were fitted and at high field were all ~0.6), however, for simplicity we label the correlation lengths as $\xi_c^{\ell=1/2,1}$. We find that the temperature dependence of the broad peak is insensitive to the exact $\ell$ value (see supplementary Fig. 1), and all $h$- and $k$-scans were carried out at $\ell = 1/2$. Third, as the $\ell \approx 1/2$ CDW component has essentially no magnetic field dependence in the narrow range 14–16 T (Fig. 2c), it is possible – by subtraction of Gaussian fits of 14 T diffraction intensities – to isolate the field and temperature dependence of the $\ell = 1$ diffraction intensity.

**Superconductivity and CDW phase competition.** By subtracting the Gaussian fit of 14 T diffraction intensity from those measured at high fields (see Supplementary Figs. 2 and 3), the temperature dependence of the $\ell = 1$ CDW component at high field is deduced. For both $B = 14.5$ and 15 T, the diffraction intensity is maximum at a finite temperature of $T \approx 25$ K (see Fig. 4). It thus demonstrates that the CDW intensity at $\ell = 1$ has a non-monotonic temperature dependence. This effect can also be analysed from in-plane intensity scans. In Fig. 5a–f, we plot the peak intensity $I$ of Gaussian fits to $k$-scans through $(0, 4 - \delta_b, \ell)$ as a function of temperature and at magnetic fields as indicated. At $\ell = 1/2$ (Fig. 5), the zero-field temperature dependence

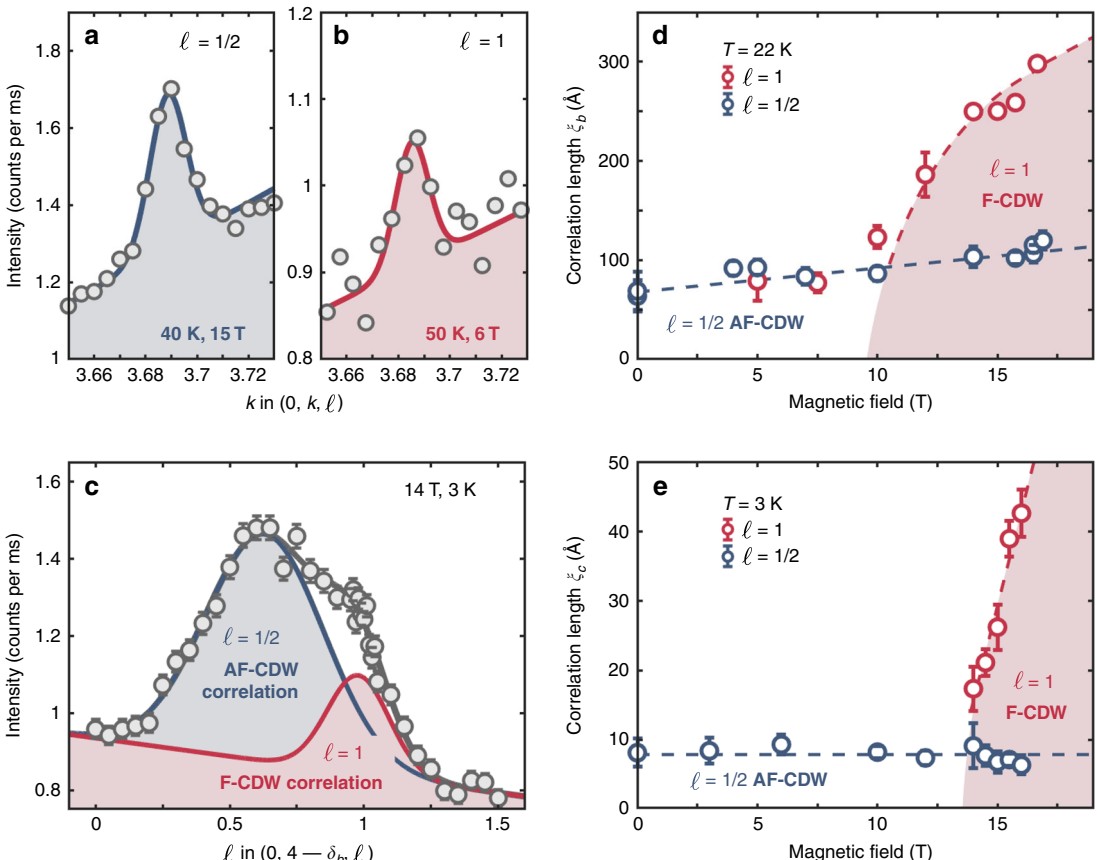

**Fig. 3 Two-component analysis of out-of-plane charge-density-wave (CDW) correlations. a, b** Raw diffraction intensities along $(0, k, 1/2)$ and $(0, k, 1)$ for temperatures and magnetic fields as indicated. **c** Raw diffraction intensity profile along the $(0, 4 - \delta_b, \ell)$ direction, fitted with two Gaussian functions on a linear background. The dark grey line is their sum. **d, e** Magnetic-field-dependent evolution of the respective correlation lengths for $\ell = 1/2$ and 1. The dashed lines and shaded area are guides to the eye. Error bars are standard deviations determined by counting statistics and Gaussian fits, respectively. Source data are provided as a Source Data file.

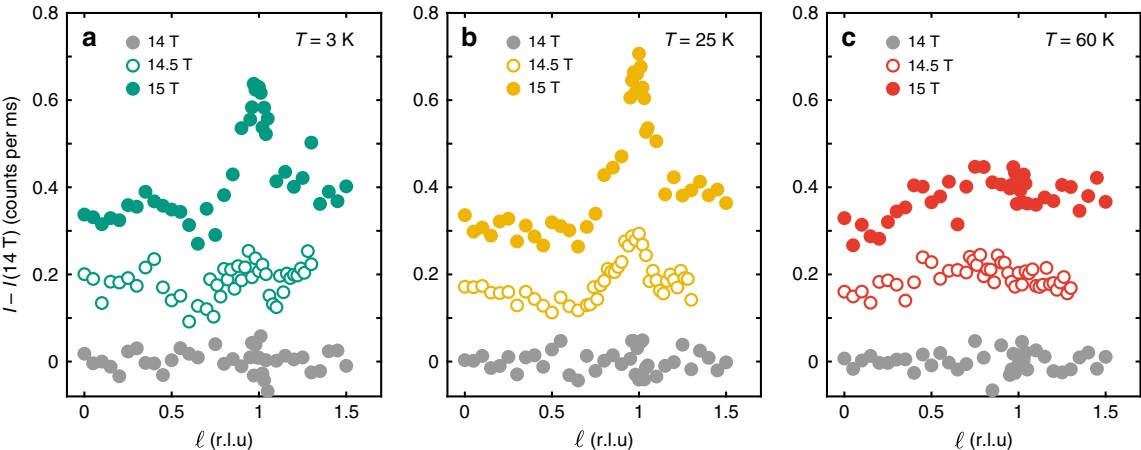

**Fig. 4 Temperature and field dependence of in- and out-of-plane diffraction intensities. a–c** $\ell$-dependent diffraction intensity after subtracting the Gaussian fitted $B = 14$ T profiles at temperatures and magnetic fields as indicated. In this fashion, the relative field effects above 14 T are extracted. Raw 14 T $\ell$-scans are shown in Supplementary Fig. 2. Data at different fields have been given an arbitrary vertical shift for the sake of presentation. Source data are provided as a Source Data file.

displays a maximum at $T = T_{\max}^{\ell=1/2} = T_c$ – consistent with previous reports[8,9]. As the magnetic field is increased, $T_{\max}^{\ell=1/2}(B)$ moves to lower temperatures. This effect is found along both $a-$ and $b-$axis directions (Fig. 5a, b).

The behaviour of the $\ell = 1$ intensity is very different. For low fields, $B \lesssim 10$ T, the field induces a relatively weak response at

$\ell = 1$. It shows a maximum at similar temperature as the $\ell = 1/2$ component (see Fig. 5d). For $10 \lesssim B \lesssim 16$ T, there is a rapid increase of $I(\ell = 1)$ at low temperature (Fig. 5c). The rapid temperature changes below 40 K associated with the crossover field may obscure any maximum due to competition with superconductivity. Then at higher fields $B \geq 14.5$ T, a maximum

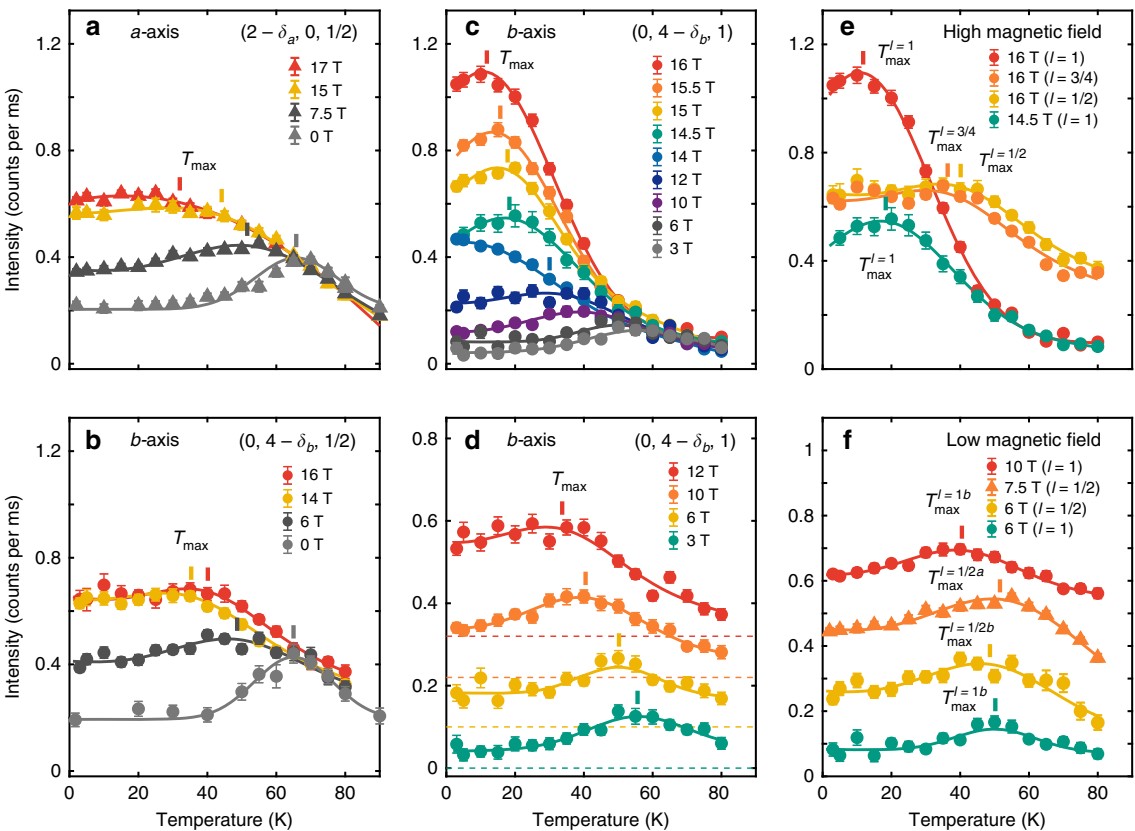

**Fig. 5 Temperature scales for phase competition. a–f** Peak intensities versus temperature measured at $\ell = 1/2$ and 1 charge-density-wave modulation vectors versus temperature. **a, b** Intensity for $(2 - \delta_a, 0, 1/2)$ and $(0, 4 - \delta_a, 1/2)$ with magnetic fields as indicated. Data in **a** and zero-field data in **b** are re-plotted from ref. [9]. Non-zero magnetic field data in **b** is $\ell \sim 0.6$ amplitudes obtained from two-Gaussian fits of $\ell$-scans. **c** Diffraction intensities at $(0, 4 - \delta_a, 1)$. **d** Same data as in **c** but zoomed on the low-field curves. For visibility, these curves have been given an arbitrary vertical shift – as indicated by the dashed base-lines. **e, f** Comparisons of $\ell = 1$ and $1/2$ temperature dependencies and with that the associated maximum temperature scales $T_{\max}$ for the high- and low- magnetic field regimes respectively. Solid lines are fits to a skewed Gaussian (see text) on a linear background. Vertical bars are the $T_{\max}$ temperature scales defined by the maximum of the second derivative of the skewed Gaussian. Error bars are standard deviations due to counting statistics. Source data are provided as a Source Data file.

again develops at $T_{\max}^{\ell=1}$ which is considerably lower than $T_{\max}^{\ell=1/2}$ for the same field. Direct comparisons of the high- and low-field diffraction intensities at $\ell = 1/2$ and $\ell = 1$ versus temperature are shown in Fig. 5e, f. To extract $T_{\max}^{\ell=1/2}$ and $T_{\max}^{\ell=1}$ in a systematic fashion where the maximum is less well defined, the diffraction intensity versus temperature curves are fitted using a skewed Gaussian function $\Gamma(T) = G(T)[1 + \text{erf}(\alpha T/\sqrt{2})]$ on a linear background (solid lines in Fig. 5). Here $G(T)$ is a Gaussian function and $\text{erf}(\alpha T)$ is the error function. The characteristic temperatures, shown in Fig. 6c versus magnetic field, are derived from the criterion $T_{\max} = \max(-d^2\Gamma/dT^2)$ – this selects either a maximum or saturation point in the diffraction intensity versus temperature. We view $T_{\max}$ as a characteristic temperature scale for competition between CDW and superconducting order.

**Correlation lengths.** Next, we turn to discuss the CDW correlation lengths. Fig. 3d, e show correlation lengths $\xi_b$ and $\xi_c$ for the lowest temperatures we measured: $T = 22$ and 3 K, respectively. The out-of-plane correlation length $\xi_c^{\ell=1/2} \sim 10$ Å is essentially magnetic-field independent while the in-plane correlation length $\xi_b^{\ell=1/2} \sim 60$ Å increases weakly. This is in strong contrast to the $\ell = 1$ order that gains a dramatic out-of-plane "coherence" above 14 T (Fig. 3e). In fact, $\xi_c^{\ell=1}$ undergoes a fourfold increase from 14 to 16 T. Prior to this change, $\xi_b^{\ell=1}$ increases significantly for field strengths above 10 T (Fig. 3d). Thus at high-fields, the correlation

length of the $\ell = 1$ order exceeds that of the $\ell = 1/2$ CDW component in both in- and out-of-plane directions. It is also interesting to discuss the in-plane correlations at the $T_{\max}$ temperature scales. For $\ell = 1/2$, the correlation length $\xi_b^{\ell=1/2}$ at $T_{\max}$, is about 20% longer than that at $T = 22$ K in the studied field range (compare Figs. 3d and 6d). The $\ell = 1$ in-plane correlations extend down to zero-magnetic-field (Fig. 6d). At $T_{\max}$, $\xi_b^{\ell=1}$ exceeds $\xi_b^{\ell=1/2}$ for $B > 5$ T (Fig. 6d), while at 22 K this is only seen at 10 T (Fig. 3d). It is of great interest to note the interplay between the observed peak intensities and the correlation lengths. Our estimates of the k-space integrated intensity for the $\ell = 1$ peak show approximately constant values versus field above 14 T (See Supplementary Fig. 4). This implies that the F-CDW develops longer range correlations with increasing field, but does not increase in amplitude or occupy a significantly larger fraction of the sample volume. The integrated intensity of AF-CDW order does not significantly increase over a range 0–16 T, which leads us to the similar conclusion for the development of AF-CDW.

## Discussion

The zero-field maximum at $T_c$ of the $\ell = 1/2$ CDW diffraction intensity has previously been the subject of different interpretations[9,24–26]. Our view is that the intensity maximum is the result of phase competition between $\ell = 1/2$ CDW order and

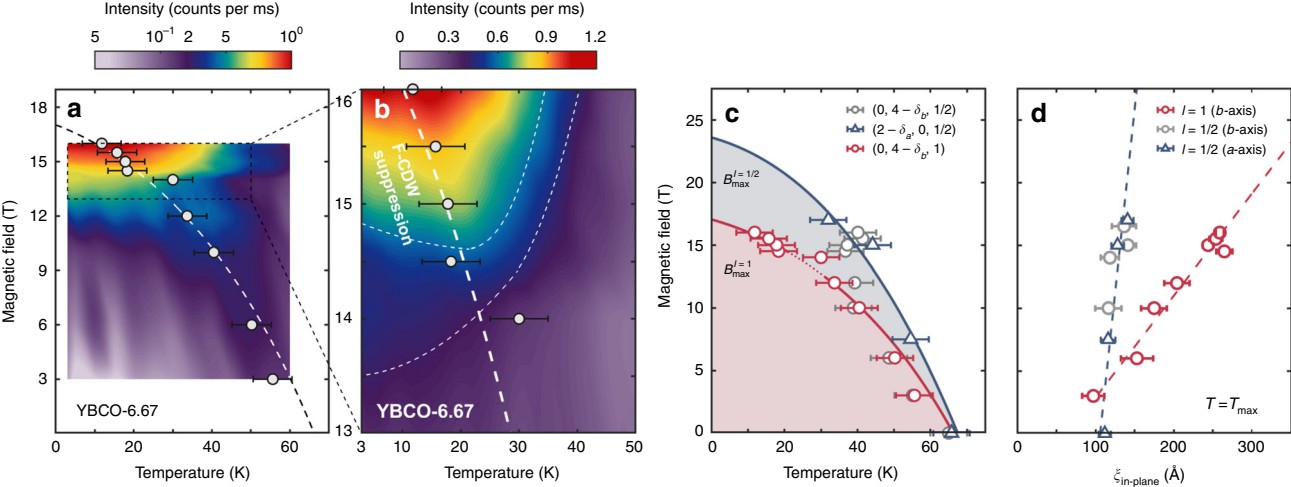

**Fig. 6 Competing temperature and magnetic field scales. a** Intensity map of $(0, 4 - \delta_b, 1)$ (logarithmic colour scale) as a function of temperature and magnetic field. The open circles indicate the temperature scale $T_{max}^{\ell=1}$ below which suppression of the F-CDW order is observed (Figs. 4a–c). The curved dashed line is a guide to the eye. **b** Intensity map (linear colour scale) focusing on the high-magnetic-field region. The thin dashed lines indicate contour lines. **c** $T_{max}$ scales inferred from the $\ell = 1/2$ (red) and $\ell = 1$ (blue) charge-density-wave, respectively. Solid lines are guides to the eye. For the $\ell = 1/2$ $T_{max}$ temperature scale, triangular and circular data points stem from data recorded along the $a$- (ref. [9]) and $b$-axis (this work) respectively. As the $T_{max}$ temperature scales indicate the onset of competition with superconductivity, these lines are labelled $B_{c2}^{\ell=1}$ and $B_{c2}^{\ell=1/2}$ in the $T \to 0$ limit. Error bars indicate the uncertainty related to the determination of the $T_{max}$ temperature scales. **d** Correlation lengths $\xi_b^{\ell=1/2}$ and $\xi_b^{\ell=1}$ at the respective $T_{max}$ temperature values. Source data are provided as a Source Data file.

superconductivity. As diffraction intensities are proportional to the square of the CDW order parameter[23,27], temperature (or field) dependence reflects the evolution of the CDW order. Upon cooling below $T_c$, the superconducting order gradually grows, resulting in a partial suppression of the CDW. The balance of this competition can be tipped in favour of the CDW order by application of an external $c$-axis magnetic field that acts to suppress the superconducting order parameter. With increasing field-strength, the $T_{max}$ temperature scale shifts to lower values (Fig. 6c) delineating a line in the magnetic field phase diagram that provides a firm lower bound for the border between superconductivity and normal state. An approximate extrapolation to the $T \to 0$ limit yields an upper critical field consistent with that inferred from most bulk measurements[28–31]. It should be stressed that there is still an ongoing debate about the exact field scale of $H_{c2}$[32]. A similar interpretation can be applied to the temperature dependence of the $\ell = 1$ response. As shown in Fig. 6c, the $\ell = 1$ maximum temperature scale $T_{max}^{\ell=1}$ is different from $T_{max}$ for $\ell = 1/2$. This implies the existence of two superconducting onset temperatures at a given applied field or for a fixed temperature two upper critical fields $B_{max}^{\ell=1/2}$ and $B_{max}^{\ell=1}$. The different competing temperature and magnetic field scales observed at $\ell = 1/2$ and $\ell = 1$ lend support to our starting hypothesis, namely that there are two different CDW order constituents. In what follows, we discuss the relationship between the CDW and the Fermi surface reconstruction.

It is interesting to consider the implications of our correlation length measurements for the electronic band structure[33,34]. Although the unfolded structure has not been resolved, the Fermi surface reconstruction (FSR) revealed by quantum oscillation[35–37] and transport[38–40] experiments is undoubtedly linked to the appearance of CDW order. It has, however, been debated whether the reconstruction is triggered by the F-CDW or AF-CDW or whether the CDW order is triggered by FSR. The fact that ARPES does not see any FSR[41] has been used as an argument for a field-induced electronic transformation, rather than the reconstruction being already present at zero-field. However, the sign change of the Hall effect persists down to as low as 5 T[42],

casting doubt on whether the high-field F-CDW order is responsible for the reconstruction. Recent detailed considerations of CDW correlations concluded that the $\ell = 1$ CDW order has about the right in-plane correlation length to explain the quantum oscillation experiments whereas the AF-CDW correlations are too short[33]. We note that at $B = 16$ T the in-plane correlation lengths of the F-CDW and AF-CDW correlations are 300 and 100 Å, respectively. Another signature of FSR is the Hall coefficient which changes sign from positive to negative on cooling[38–40,43]. It has been demonstrated that for YBCO $p \approx 0.12$, the characteristic sign-changing temperature $T_B = 66$ $K \simeq T_c$ is essentially independent of magnetic field[38]. The sign change in the Hall effect persists down to much lower fields, $B = 5$ T, than the quantum oscillations mentioned above. Thus it is interesting to consider whether the F-CDW correlations might be involved in the FSR at lower fields. We have previously demonstrated[14] that for high magnetic fields, $B = 16.5$ T, $\xi_b^{\ell=1} > \xi_b^{\ell=1/2}$ for $T \approx 66$ $K \approx T_B$. We can see from Fig. 6d that this condition persists at $T_{max}$ as $B \to 0$ and $T_{max} \to T_B$. In fact, we find that $\xi_b^{\ell=1}$ is longer than $\xi_b^{\ell=1/2}$ down to 5 T where $T_{max} \approx 55$ K. Thus, F-CDW correlations with $\xi_b^{\ell=1} \geq 100$ Å $\approx \xi_b^{\ell=1/2}$ exist for temperatures $T \lesssim T_B$ and to lowest measured magnetic fields. Considering the $(B, T)$ phase space, both the F-CDW and AF-CDW correlations could be associated with the Fermi surface reconstruction inferred from Hall effect experiments. However, in both cases the correlation lengths are shorter than the expected threshold for quantum oscillations to be observed. Therefore, either the two-dimensionality of the $CuO_2$ enables the electronic structure to fold also for intermediate correlation lengths or the Fermi surface reconstruction has an entirely different causality and charge order is consequence rather than a trigger.

A central part of this work is the experimental probing of $\ell = 1$ in-plane correlations as a function of magnetic field and temperature. With this, the interaction between $\ell = 1$ CDW order and superconductivity can be discussed. The $\ell = 1$ order displays different competing field and temperature scales, compared to the extensively studied $\ell = 1/2$ order[8,9,22,23,44,45]. The $T_{max}^{\ell=1}$ and

$T_{\max}^{\ell=1/2}$ temperature scales indicate the point where super-conductivity is strong enough to partially suppress the respective CDW orders. In presence of large enough magnetic fields $B \geq 14$ T, we find that $T_{\max}^{\ell=1} < T_{\max}^{\ell=1/2}$ (Fig. 5e). Extrapolating $T_{\max}^{\ell=1} \to 0$, a field scale significantly smaller than $B_{c2}$, inferred from transport measurements[28,29,40], is reached. In contrast, $T_{\max}^{\ell=1/2}$ may track $B_{c2}$ at high fields. This suggests that super-conductivity occuring in regions with $\ell = 1$ CDW order is weaker than in $\ell = 1/2$ regions. This is especially the case when the $\ell = 1$ correlations develop coherence along the $c$-axis. The emergence of ferro-coupled long-range $\ell = 1$ CDW order, therefore, induces a fragile state of superconductivity. In the presence of quenched disorder, such a fragile state of superconductivity has, in fact, been predicted theoretically[17]. Disorder may therefore play an important role for the organisation of the observed phase diagram[17,18] and the interaction between CDW order and superconductivity. The exact role of different types of disorder (oxygen-chain vacancies, impurities and vortices) on the AF-CDW order and the fragile superconducting state should be clarified in future experiments. The regime with competition induced weakening of superconductivity has been the setting for predictions of pair-density-wave formations[3,46–50]. The existence of such a state should manifest itself through a $\mathbf{Q}_{PDW} = \mathbf{Q}_{CDW}/2$ modulation. Although STM studies have reported evidence for PDW order in the halos of vortices[51], there are no x-ray diffraction results supporting the existence of a PDW order in YBCO.

Finally, we comment on the $S$-shape of $B_{DOS}$ versus temperature reported by density-of-states sensitive probes (specific heat, thermal conductivity, and NMR)[52]. For $B > B_{DOS}$, the density-of-states is magnetic-field independent and the $S$-shape, refers to an effect where $B_{DOS}$, in addition to standard Bardeen-Cooper-Schrieffer (BCS) behaviour, displays an sudden increase in the $T \to 0$ limit. Our results do not exclude the possibility that $B_{\max}^{\ell=1}$ would increase steeply in the $T \to 0$ limit. In fact, it is possible that the $\ell = 1/2$ and $\ell = 1$ superconducting flavours couple in the $T \to 0$ limit to form a uniform condensate. If so, $B_{\max}^{\ell=1}$ and $B_{\max}^{\ell=1/2}$ should merge for $T = 0$. This would lead $B_{\max}^{\ell=1}$ to rise in the low-temperature limit and give it an $S$-shaped dependence reminiscent of that reported by thermodynamic and spin susceptibility probes of $B_{c2}$[52].

## Methods

**Experimental details.** A high quality single crystal of Y Ba$_2$Cu$_3$O$_{6.67}$ ($T_c = 67$ K) with ortho-VIII oxygen ordering was grown and detwinned as described in ref. [53]. Hard x-ray (100 keV) diffraction experiments were carried out, with a horizontal 17 T magnet[54], at PETRA III's P07 triple-axis diffractometer at DESY (Hamburg, Germany). The YBCO sample was mounted with the magnetic field along the $c$-axis direction giving access to the $\mathbf{Q} = (h, 0, \ell)$ scattering plane. The setup is identical to that described in ref. [14], with the exception of an improved sample cooling power allowing a base temperature of $T \approx 3$ K to be reached. Scattering vectors are specified in orthorhombic notation with reciprocal lattice units (r.l.u) ($2\pi/a$, $2\pi/b$, $2\pi/c$) where $a \simeq 3.82$ Å, $b \simeq 3.87$ Å, and $c \simeq 11.7$ Å.

## Data availability

All experimental data are available upon request to the corresponding authors. The source data underlying Figs. 2–6, and Supplementary Figs. 1–4 are provided as a Source Data file.

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

## Acknowledgements

We acknowledge informative discussions with Steve Kivelson. This work was supported by the EPSRC (grant numbers EP/R011141/ & EP/J016977/1), the U.S. Department of Energy (DOE), under Contract No. DE-AC02-98CH10886, the Wolfson Foundation, the Royal Society, the Office of Basic Energy Sciences, Division of Materials Science and Engineering, the Danish Agency for Science, Technology and Innovation under DAN-SCATT, the Leverhulme Trust, and the Swiss National Science Foundation. Parts of this research were carried out at beamline P07 at DESY, a member of Helmholtz Association HGF.

## Author contributions

R.L., D.A.B. and W.N.H. grew and characterised the $YBa_2Cu_3O_{6.67}$ single crystal. E.B., O.I., A.T.H., N.B.C., M.H., E.M.F., O.G., U.R., M.v.Z., S.M.H. and J. Chang executed the x-ray diffraction experiments. J. Choi, O.I., S.G., S.M.H. and J. Chang analysed the data. All authors contributed to the manuscript. J. Choi and O.I. equally contributed to this work.

## Competing interests

The authors declare no competing interests.
