## [Peer Review File · Nature Communications]

Reviewers' comments:

Reviewer #1 (Remarks to the Author):

The authors present a detailed x-ray scattering study of CDW order in YBCO(6.67) as a function of temperature and magnetic field. In particular, they distinguish between two types of CDW "order", one with AF correlations along the c-axis and the other with F correlations along c. They characterize the different behaviors of the correlation lengths and intensities of these two orders temperature and field. They relate these behaviors to 1) competition with the superconducting order, and 2) relevance to quantum oscillations (observed at higher fields).

This is a careful and thoughtful study on a topic that continues to be of considerable interest. I recommend that the paper be accepted after the authors have had a chance to respond to the comments below.

Comments:

1) Suggestion: p.3, line 113, change "separated by planes containing" to "separated by layers containing"

2) p. 6, lines 276-278, comparing an extrapolated upper critical field with "that inferred from bulk measurements", and p. 7, final paragraph (lines 378-392).

It would be appropriate in one of these places to include a comment and reference to F. Yu et al., PNAS 113, 12667 (2016). In that paper, torque magnetometry measurements provide evidence for weak diamagnetism extending to > 40 T. At low temperature, there is evidence for a transition at ~ 25 T from a phase with a large magnetic hysteresis to one with a much small hysteresis and extending to ~ 39 T. There is a strong overlap with the "S-shape" behavior reported in Ref. 45.

3) p. 7. On line 325, the parameter T_H is introduced. In the following line, and also in line 329, a different parameter, T_B , is mentioned without explanation. Should T_B be T_H ? If not, please explain what it represents.

4) p. 6, line 291, "Fermi surface reconstruction". The authors use this phrase in a manner that is commonly used in this field; however, I just want to note that it is somewhat misleading. Quantum oscillation studies provide evidence for a quasiparticle response on at least some arcs of Fermi surface. Because the inferred effective pockets are inconsistent with the Fermi surface calculated from models that ignore relevant interactions, the pockets are often attributed to "Fermi surface reconstruction". The problem I have with this usage is that there is no direct experimental evidence for the "un-reconstructed" Fermi surface in this compound at this doping for any temperature or field. I am not trying to suggest that the CDW order is not relevant to the pockets inferred from QO studies, but rather that there is no evidence that the CDW order is in any way "conventional". I don't expect the authors will make any changes on this point, but I wanted to share my opinion.

Reviewer #2 (Remarks to the Author):

In the submitted manuscript (NCOMMS-19-30883), Choi *et al.* present a comprehensive x-ray scattering study with a magnetic field up to 17 Tesla on ortho-VIII YBCO. Through the comparison between AF-CDW ($l \sim 1/2$) and F-CDW ($l \sim 1$) CDW orders and under the magnetic field and corresponding temperature dependence, the authors found that a response of each CDW intertwined with the SC phase is different, indicating an inhomogeneous superconducting state exists at CuO₂ plans. Based on these findings, they argue that two types of superconducting states are co-existing; in particular, a weak or fragile SC may be related to the putative pair-density-wave order.

These findings and corresponding discussions are quite interesting and potentially important. In particular, the study of high magnetic field x-ray scattering gives a good clue why the 3D CDW order in YBCO is more intrinsic and ideal. Therefore, I would be supportive of this work. However, before my recommendation, a few things, which are described below, should be addressed.

1. It is not clear whether an origin of F-CDW order is related to the AF-one or not.
2. Figure 2 shows the inhomogeneous CDW orders. Does the corresponding SC spatial distribution overlap with the two orders? Also, I strongly recommend that authors should add a schematic figure like Fig. 2a for SC distribution.
3. Even though Fig. 4b tried to show the only AF order behavior, it is still an effect from the tail of 3D (as like the precursor). How could you remove this effect?
4. Since the T_{\max} at $l = 1$ is lower than that at $l = 1/2$, as I understand, such argument of the fragility is working. However, it would be possible to explain – the F-CDW order is actively suppressing SC because the order is even stronger, instead of the fragility. How do you distinguish this?
5. Would you clarify which CDW / SC (i.e., 3D/fragile or the other) type has more disorder?

Minor comments

- The correlation length along l -direction in the low field range is obtained by fitting with two Gaussian peak, but high field >14 T is subtracted 14 T as background. Why not using a consistent process? If consistent, how to change the result? Should be consistent.
- The low magnetic-field range data is fitted with two Gaussian peak, but as the author pointed out, that $l \sim 1/2$ CDW peak is not really maximized at $l \sim 1/2$, but the l -value approximately equal to 0.6. This uncertainty of the center would induce error to the fitting result of both peaks intensity and fitted width. How do the authors estimate this error?
- In the line 385, 389, and 390, it seems the authors refer to H_{\max} instead of T_{\max} , are they typos?
- The authors show the integrated intensity is consistent for AF-CDW in a field range from 0-16 T, and F-CDW in the field range over 14 T. However, as it has been discussed in the paper, both CDW could be related to the disorder in the sample, which can be changed by the strength of magnetic field. And as it's well known, the integrated intensity of AF-

CDW at zero field is changing with temperature, how to understand the consistent behavior with the magnetic field.

- In the line 200-202, the authors stated: “At the ‘crossover field’ $B = 14$ T, I ($l = 1$) increases monotonically with decreasing temperature, i.e., no maximum is observed.” Is there any explanation about the reason why it is monotonically changed in this condition?
- According to Fig. 5c, the condition $T_{\max}^{l=1} < T_{\max}^{l=1/2}$ already satisfied at low field range (say 5 T). Does it mean the proposed new superconducting states show up in this low field region? The discussion about the region where this “fragile” state is a little obscure.

Reviewer #1 (Remarks to the Author):

The authors present a detailed x-ray scattering study of CDW order in YBCO(6.67) as a function of temperature and magnetic field. In particular, they distinguish between two types of CDW “order”, one with AF correlations along the c-axis and the other with F correlations along c. They characterize the different behaviors of the correlation lengths and intensities of these two orders temperature and field. They relate these behaviors to 1) competition with the superconducting order, and 2) relevance to quantum oscillations (observed at higher fields).

This is a careful and thoughtful study on a topic that continues to be of considerable interest. I recommend that the paper be accepted after the authors have had a chance to respond to the comments below.

We thank Reviewer 1 for his/her summary and supportive recommendation. Below, we respond in a point-by-point fashion to the suggestions for improvements.

Comments:

1) Suggestion: p.3, line 113, change “separated by planes containing” to “separated by layers containing”

We amended the manuscript according to this suggestion (See Page 2 line 117 in the manuscript revised).

2) p. 6, lines 276-278, comparing an extrapolated upper critical field with “that inferred from bulk measurements”, and p. 7, final paragraph (lines 378-392). It would be appropriate in one of these places to include a comment and reference to F. Yu et al., PNAS 113, 12667 (2016). In that paper, torque magnetometry measurements provide evidence for weak diamagnetism extending to > 40 T. At low temperature, there is evidence for a transition at ~ 25 T from a phase with a large magnetic hysteresis to one with a much small hysteresis and extending to ~ 39 T. There is a strong overlap with the “S-shape” behavior reported in Ref. 45.

We have included the mentioned reference (F. Yu et al.) and stressed there is an ongoing debate about the exact field scale of H_{c2} (See page 5 line 285-286)

3) p. 7. On line 325, the parameter T_H is introduced. In the following line, and also in line 329, a different parameter, T_B , is mentioned without explanation. Should T_B be T_H ? If not, please explain what it represents.

Thanks for pointing out this inconsistency. We now use T_B consistently throughout the manuscript (Please see page 6 line 334).

4) p. 6, line 291, “Fermi surface reconstruction”. The authors use this phrase in a manner that is commonly used in this field; however, I just want to note that it is somewhat misleading. Quantum oscillation studies provide evidence for a quasiparticle response on at least some arcs of Fermi surface. Because the inferred effective pockets are inconsistent with the Fermi surface calculated from models that ignore relevant interactions, the pockets are often attributed to “Fermi surface reconstruction”. The problem I have with this usage is that there is no direct experimental evidence for the “un-reconstructed” Fermi surface in this compound at this doping for any temperature or field. I am not trying to suggest that the CDW order is not relevant to the pockets inferred from QO studies, but rather that there is no evidence that the CDW order is in any way “conventional”. I don't expect the authors will make any changes on this point, but I wanted to share my opinion.

We agree that the unconstructed Fermi surface has not been probed. The existence of the pseudogap that forms Fermi arcs certainly drives the entire system beyond any conventional concepts. To accommodate part of this comment, we now mention directly in the text that the unfolded structure has (also) not been resolved (See page 5 line 302-304)

Reviewer #2

In the submitted manuscript (NCOMMS-19-30833), Choi et al. present a comprehensive x-ray scattering study with a magnetic field up to 17 Tesla on ortho-VII YBCO. Through the comparison between AF-CDW ($l \sim 1/2$) and F-CDW ($l \sim 1$) orders and under the magnetic field and corresponding temperature dependence, the authors found that a response of each CDW field intertwined with the SC phase is different, indicating an inhomogeneous superconducting state exists at CuO₂ planes. Based on these findings they argue that two types of superconducting states are co-existing; in particular, a weak or fragile SC may be related to the putative pair-density-wave order.

These findings and corresponding discussions are quite interesting and potentially important. In particular, the study of high magnetic field x-ray scattering gives a good clue why the 3D CDW order in YBCO is more intrinsic and ideal. Therefore, I would be supportive of this work. However, before my recommendation, a few things, which are described below, should be addressed.

We also wish to thank Reviewer 2 for his/her summary and constructive suggestions:

1. It is not clear whether an origin of F-CDW order is related to the AF-one or not.

This is a good point by the reviewer. It should be more clearly articulated in the manuscript text that the AF- and F-CDW orders are linked. Exactly how is part of the problem to understand YBCO in magnetic fields. We now state in the introduction (page 1 line 39-40)

“As a function of both magnetic field and uniaxial stress, two intimately related CDW ordering tendencies have been realized.”

We have also added a real space picture of the ordering in the new Figure 1.

2. Figure 2 shows the inhomogeneous CDW orders. Does the corresponding SC spatial distribution overlap with the two orders? Also, I strongly recommend that authors should add a schematic figure like Fig. 2a for SC distribution.

There is consensus that superconductivity in YBCO-6.67 is a bulk effect. As such it should coexist with both F-CDW and AF-CDW orderings, yes.

The general thesis of the manuscript is that there is an inhomogeneous superconducting state and the superconductivity responds differently to the F-CDW and AF-CDW orders. From the present experiment we cannot determine the spatial distribution of the superconducting gap. However, we can say that the superconductivity responds differently to the F-CDW and AF-CDW region. We have added a new panel to Fig. 2 (now Fig 1) to show what the disordered CDW might look like at the microscopic level. We expect the gap to be smaller in the F-CDW regions.

3. Even though Fig. 4b tries to show the only AF order behavior, it is still an effect from the tail of 3D (as like the precursor). How could you remove this effect?

In a general sense, the AF-CDW order is a precursor of the F-CDW, as they have essentially identical basal plane wavevectors. We believe that the AF CDW order is most likely present because it is nucleated by quenched disorder. In another system, it might be removed by removing all the disorder. However, this is not possible in YBCO because the oxygen chains, which create the disorder, are used to dope this system and cannot be removed. In the caption of Fig. 4, it is stated that data in Fig. 4b is analysed using two-Gaussian fits to I-scans. This is disentangling the AF- and F-CDW ordering components.

4. Since the T_{\max} at $l = 1$ is lower than that at $l = 1/2$, as I understand, such argument of the fragility is working. However, it would be possible to explain – the F-CDW order is actively suppressing SC because the order is even stronger, instead of the fragility. How do you distinguish this?

This is exactly our view that the fragile superconducting state emerges due to stronger competition with the F-CDW order. The notion of fragile superconductivity as discussed in Ref. [17] is that the regions where superconductivity is suppressed can Josephson couple together, yield phase coherence and thus be observed – as in this experiment. The fragile state is a specific case of the

reviewer's proposal and as such cannot be distinguished.

5. Would you clarify which CDW / SC (i.e. 3D/fragile or the other) type has more disorder?

The disorder in the system comes from two possible sources. (1) quenched disorder; (2) vortices inserted into the system with magnetic field. We believe the disorder is primarily due to quenched disorder because the CDW is strongly disordered. Also application of a magnetic field tends to lead to a more ordered system. We have added a note in the introduction section: "The introduction of vortices with field leads to a more ordered CDW. This suggests disorder associated with vortices is of secondary importance." (See page 1 line 54-57)

Theoretically, disorder has been used to predict both pinning of AF-CDW order and to be the source of fragile superconductivity. Reference to such theoretical work is made within our manuscript. However, based on our current experimental data, it is impossible to resolve completely exactly how different types of disorder influence the different CDW and SC phases. We have added as a sentence to the discussion stating this fact and that future experiments should address this point (See page 7 line 377-380).

Minor comments:

1. The correlation length along l -direction in the low field range is obtained by fitting with two Gaussian peak, but high field > 14 T is subtracted 14 T as background. Why not using a consistent process? If consistent, how to change the result? Should be consistent.

Subtraction of 14-T background is only applied in Fig. 3 and in Fig. S2 to demonstrate more clearly the suppression of F-CDW order at low temperature. Except for these two figures, we consistently employed the "two-Gaussian" model and a single Gaussian function for analysis of l -scans and k -scans, respectively. These three different ways of analyzing the data lead to consistent conclusions. We now stress, in the text, that the data analysis is carried out using three separately independent methodologies that lead to consistent results (See page 3 line 159)

2. The low magnetic-field range data is fitted with two Gaussian peak, but as the author pointed out, that $l \sim 1/2$ CDW peak is not really maximized at $l \sim 1/2$, but the l -value approximately equal to 0.6. This uncertainty of the center would induce error to the fitting result of both peaks intensity and fitted width. How do the authors estimate this error?

Here the reviewer presumably assumes that the AF-CDW peak position was fixed to $l=0.6$ and that this would artificially reduce the error bar on the width and amplitude. In fact, we did keep the AF-CDW peak position as a free fitting parameter and for all fits it had a value around 0.6. As such, the error bars are defined as the standard Gaussian deviations. In the revised manuscript, we now state directly that the AF-CDW peak position is an open fit parameter (See page 3 line 170). Note that the peak is not expected to be at $l=1/2$ because of the structure factor of the CDW structure. (See the ref. [14]).

3. In the line 385, 389, and 390, it seems the authors refer to H_{\max} instead of T_{\max} , are they typos?

Yes, indeed it should be B_{\max} instead of T_{\max} . We corrected accordingly (See page 7 line 397, 401, 402).

4. The authors show the integrated intensity is consistent for AF-CDW in a field range from 0-16 T, and F-CDW in the field range over 14 T. However, as it has been discussed in the paper, both CDW could be related to the disorder in the sample, which can be changed by the strength of magnetic field. And as it's well known, the integrated intensity of AF-CDW at zero field is changing with temperature, how to understand the consistent behavior with magnetic field.

This comment refers to supplementary figure S3 that is briefly mentioned in the main text. It is indeed interesting that the integrated intensities of the AF-CDW order remain almost insensitive to magnetic field that induces disorder in terms of vortices. As mentioned in an earlier reply, vortices are theoretically expected to pin the AF-CDW order. We stress that the scattering amplitude – often linked to the order parameter – and the in-plane correlation length both increase with magnetic field strength. A constant integrated intensity is therefore not necessarily inconsistent with disorder

scenarios. It shows that the disorder introduced by the vortices has no significant effect. The quenched disorder is dominant. Finally, magnetic field- and temperature-dependence are not necessarily linked.

5. In the line 200-202, the authors stated: “At the ‘crossover field’ $B = 14$ T, $I(l = 1)$ increases monotonically with decreasing temperature, i.e., no maximum is observed.” Is there any explanation about the reason why it is monotonically changed in this condition?

At the cross-over field of 14 Tesla, a rapid variation of the F-CDW order parameter along the c-axis with temperature below 40K is observed. This may obscure the observation of a maximum. We have added a sentence to explain this (See page 4 line 206-209).

6. According to Fig. 5c, the condition $T_{max}^{l=1} < T_{max}^{l=1/2}$ already satisfied at low field range (say 5T). Does it mean the proposed new superconducting states show up in this low field region? The discussion about the region where this “fragile” state is a little obscure.

This point 6 contains a question and a comment.

The question about the new superconducting state is based on the assumption that $T_{max}^{l=1} < T_{max}^{l=1/2}$. We understand the temptation to make this conclusion based on the plotted data. During the preparation of the manuscript, we considered this very carefully but within the statistical error bars, our data is not conclusive on this point. It is however an interesting point to settle and we have already applied for more beam time, exactly to address this question. We have amended the text to reflect this fact (See page 7 line 361-362). For the comment about obscure discussion of the “fragile” state, we are thinking that the reviewer is referring again to the fact that both AF-CDW order and fragile superconductivity are theoretically predicted to be promoted by disorder. We addressed this though point 5 of the main suggestions. In addition to this, we have shortened slightly other parts of this discussion.

REVIEWERS' COMMENTS:

Reviewer #2 (Remarks to the Author):

In this revised version, the authors have tried to address all my, as well as referee 1, comments/questions, as much as possible. I appreciate these efforts. Now, the manuscript gets better to read for a general reader. Although there are still a few unclear parts in the revised manuscript, I understand that it is hard to address the parts clearly at this moment. In this sense, I would like to recommend the publication.

Point-by-point reply to reviewer #2

In this revised version, the authors have tried to address all my, as well as referee 1, comments/questions, as much as possible. I appreciate these efforts. Now, the manuscript gets better to read for a general reader. Although there are still a few unclear parts in the revised manuscript. I understand that it is hard to address the parts clearly at this moment. In this sense, I would like to recommend the publication.

We thank the reviewer 2 for his or her second review and recommendation for publication in Nature Communications.